# The Role of Internalized Transphobia, Loneliness, and Social Support in the Psychological Well-Being of a Group of Italian Transgender and Gender Non-Conforming Youths

**DOI:** 10.3390/healthcare10112282

**Published:** 2022-11-14

**Authors:** Maria Garro, Cinzia Novara, Gaetano Di Napoli, Cristiano Scandurra, Vincenzo Bochicchio, Gioacchino Lavanco

**Affiliations:** 1Department of Psychology, Educational Science and Human Movement, University of Palermo, 90128 Palermo, Italy; 2Department of Neurosciences, Reproductive Sciences and Dentistry, University of Naples Federico II, 80138 Naples, Italy; 3Department of Humanistic Studies, University of Calabria, 87036 Rende, Italy

**Keywords:** internalized transphobia, loneliness, transgender

## Abstract

Although transgender and gender non-conforming (TGNC) youth represent a highly resilient community capable of successfully overcoming adverse life circumstances, they still face social stigma that negatively impacts their health, being at risk of developing negative feelings toward their own TGNC identity (i.e., internalized transphobia). A poorly investigated dimension in TGNC health research is perceived loneliness. Thus, within the minority stress theory, the present study aimed to investigate the mediating role of loneliness and the moderating role of social support in the relationship between internalized transphobia and psychological well-being among 79 Italian TGNC youths aged 18 to 30-years-old who participated in an online survey. The main results show that loneliness partially mediated the relationship between internalized transphobia and psychological well-being. In addition, social support emerged as a significant moderator, as the impact of internalized transphobia on psychological well-being decreased at moderate and high levels of social support, but not at low levels. The findings have significant implications for clinical practice and psychosocial interventions to reduce the impact of internalized stigma and stress on psychological health.

## 1. Introduction

Transgender and gender non-conforming (TGNC) individuals include all people whose gender identity does not fully align with their sex assigned at birth and who may or may not adhere to a binary conception of gender [1]. Therefore, TGNC individuals include those who were assigned male (AMAB) or female (AFAB) at birth and who identify themselves as men or women, respectively, or do not identify as men or women (i.e., non-binary), or identify with both genders or at some point along the gender continuum [2,3]. Not all TGNC individuals need to undergo surgery to affirm their perceived gender (Gender Affirmation Surgery) or undergo hormonal treatments to align their body with their perceived gender identity [4].

Despite the extreme diversity of the population, TGNC individuals have in common the belonging to a social minority group, since gender expression, roles, and identity are shaped by prescribed heteronormative cultural rules. These rules can also ensure that TGNC people can represent the symbol of ‘deviation’ from normative gender expectations, since the expectations related to gender differences are idealized and socially shared. Understanding, therefore, the threats to the well-being of TGNC people means taking into account the ways in which these expectations come together in the different contexts of daily life (work, health, sports, education) [5,6,7]. For example, the younger TGNC population appears to be more vulnerable than the adult population [8]. In fact, students experience forms of transphobic bullying at higher rates than bullying that does not have this connotation. This implies the possibility for the young TGNC population to understand whether they have the freedom to be able to express their identity without incurring risks of any kind; it is in this way, therefore, that heteronormativity becomes oppressive, as understood as just and natural by the majority group [9]. These are the reasons why TGNC individuals, like other minority individuals, experience chronic stress related to ongoing stigma, discrimination, and prejudice against their identity, in addition to the general stressors that apply to the general population. Some examples include exclusion, peer rejection, and transphobic micro-aggressions [10,11,12].

One of the most useful theoretical frameworks for understanding the life experiences of TGNC individuals in relation to stigma, stress, and health is the minority stress theory (MST) [13,14]. Originally, the MST was applied to lesbian, gay, and bisexual (LGB) populations. More recently, TGNC people have also been included [15,16,17,18,19,20], providing an essential perspective on the relationship between experiences of discrimination and well-being for TGNC individuals. Indeed, the MST suggests that distal stressors (i.e., discrimination, rejection, and victimization) resulting from anti-TGNC stigma and proximal stressors (i.e., internalized transphobia, negative future expectations, and perceived stigma) have a negative and direct impact on individual health [15,21,22]. However, the stressor most strongly associated with health problems is internalized transphobia, i.e., concern about one’s TGNC identity due to internalization of society’s gender normative expectations, which may be directed against oneself (vertical internalized transphobia) and/or other TGNC individuals (horizontal internalized transphobia) [16,23]. Internalized transphobia in TGNC youth may exacerbate clinically significant post-traumatic stress disorder symptoms [24,25] and, in some cases, possible suicidal thoughts [26,27] or non-suicidal self-injury [28].

The consequences of pervasive stress associated with TGNC identity can be significant. This is because social and internalized stigma is associated with higher psychological distress and impedes the path to well-being and self-acceptance [16,29,30]. In this regard, risk factors for the well-being of TGNC youth appear to be primarily stigma and the resulting potential loss of social support [31]. The latter seems to implement the risk of experimenting with potential depression problems and feelings of loneliness [32,33]. TGNC people who engage in stigma, prejudice, and discrimination may compromise their access to social support because internalized stigma creates feelings of shame and guilt.

Loneliness is a subjective feeling of discrepancy between desired and adequate levels of social contact [34]. Perceived loneliness peaks during adolescence, which is characterized by the rapid development of biological and social relationships [35], as well as identity development, in which gender identity finds its place. Stigma (internalized and social) is associated with loneliness due to feared negative interactions [36]. Very little is known about the perceived loneliness of TGNC individuals, especially in the Italian environment, which is the context of the current study and that, as reported by previous research, is not very supportive of TGNC people, although they are becoming more visible [37,38,39].

Therefore, according to several authors [15,18,19], it seems necessary to identify possible factors able to mediate or mitigate the interaction between internalized transphobia and psychological well-being. In the context of this study, we consider loneliness and social support in the relationship between internalized transphobia and psychological well-being among a group of Italian TGNC youths. Indeed, it seems plausible to assume that TGNC individuals with high levels of internalized transphobia have a harder time belonging to a community and feel lonely because of a negative self-perception due to this stressor.

## 2. The Current Study

In the context of the MST, the present study aimed to assess the relationships of some subjective (i.e., internalized transphobia and loneliness) and protective (i.e., social support) factors with psychological well-being in a group of Italian TGNC youths.

Specifically, we hypothesized that: (1) internalized transphobia is positively associated with loneliness; (2) both internalized transphobia and loneliness are negatively associated with psychological well-being; (3) loneliness mediates the relationship between internalized transphobia and psychological well-being; and (4) perceived social support buffers the negative effect that internalized transphobia has on psychological well-being.

The hypothesized moderated mediation model is shown in Figure 1.

## 3. Methods

### 3.1. Participants

The current study analyzed data from 79 Italian TGNC youths who had participated in an online survey. Participants ranged in age from 18 to 30-years-old (*M* = 23.73, *SD* = 3.59). Among them, 22 were AMAB and 57 were AFAB. Participants were eligible to participate in the survey if they self-identified with a TGNC identity, were between 18 and 30-years-old, and lived in Italy. Socio-demographic characteristics are shown in Table 1.

### 3.2. Procedures

Participants were recruited through a snowballing process by introducing the research project in an academic context and simultaneously disseminating a link to access an online questionnaire. The link was posted on the websites of social and educational institutions (e.g., the official website of the University of Palermo) and self-help associations (local LGBTQI+ communities, e.g., ArciGay Palermo).

In addition, the link to the survey was posted in Italian TGNC social media groups on Facebook and Instagram, and national stakeholders were contacted to ask them to share the survey with potentially interested TGNC individuals. The following announcement was posted: “A study is currently underway in Italy to investigate the role of some psychological factors in the well-being of TGNC individuals. We are looking for people between 18 and 30 years of age whose gender identity differs from the sex assigned at birth and who live in Italy”.

The objectives and voluntary nature of the study were explained to participants on the first page, and informed consent was obtained by asking them to fill out a form on the above platform.

The project was reviewed by the Ethics Committee for Psychological Research of the University of Palermo (protocol number 10/2020). The data collection procedure fully complied with the Ethical Research Code of the Italian Association of Psychology and the ethical recommendations of the Declaration of Helsinki, as well as with the standards of the American Psychological Association for the treatment of human volunteers.

### 3.3. Measures

Socio-demographic information. Socio-demographic variables included age, gender assigned at birth (AMAB vs. AFAB), actual gender identity (binary vs. non-binary), education level (high school or less vs. college or more), ethnicity (Caucasian vs. non-Caucasian), stable relationship (yes vs. no), community size (urban vs. non-urban), and religious education (yes vs. no). In terms of actual gender identity, participants were categorized as binary if they described themselves as women, men, transwomen, or transmen, and non-binary if they described themselves as genderqueer, bigender, or another gender identity that fell outside of a binary conception of gender.

Internalized transphobia. Internalized transphobia was assessed using the “Internalized Transphobia” subscale of the Gender Minority Stress and Resilience Scale [15,40,41]. This is an eight-item scale that assesses shame toward one’s TGNC identity caused by internalizing negative societal views about TGNC identities. An example item is “Because of my gender identity or expression, I feel like an outcast.” Response options range from 1 (strongly disagree) to 5 (strongly agree), and the total score is calculated by summing the scores of each item, thus ranging from 8 to 40. The alpha coefficient for the current sample was 0.91.

Loneliness. Loneliness was assessed with the Loneliness and Aloneness Scale (LACA) [42,43]. The LACA is a20-item questionnaire assessing perceived loneliness. An example item is “I feel isolated from others.” Each item can be answered on a four-point scale ranging from (1) never to (4) often. The alpha coefficient for the current sample was 0.71.

Perceived social support. Perceived social support was assessed through the Multidimensional Scale of Perceived Social Support (MSPSS) [44,45], a 12-item scale that measures perceived social support from family (e.g., “My family really tries to help me”), friends (e.g., “My friends really try to help me”), and significant others (e.g., “There is a special person who is around when I am in need”). Response options range from 1 (very strongly disagree) to 7 (very strongly agree). The total score is calculated by summing the scores of each item and dividing the score obtained by the number of items. For simplicity, in the current study, we used the total score, with an alpha coefficient of 0.90.

Psychological well-being. Psychological well-being was assessed using the Warwick–Edinburgh Mental Well-being Scale (WEMWBS) [46,47]. The questionnaire consists of 14 items measuring psychological well-being on a five-point Likert scale, ranging from 1 (never) to 5 (always). The items are phrased in a positive sense about psychological well-being. A higher score corresponds to a higher level of well-being. The alpha coefficient for the current sample was 0.93.

### 3.4. Statistical Analyses

Statistical analyses were performed using the Statistical Package for the Social Sciences (SPSS 27).

First, descriptive statistics of participants and bivariate correlations between internalized transphobia, loneliness, social support, and psychological well-being were calculated.

Second, a moderated mediation model analysis was conducted to test the direct and mediating effects of internalized transphobia and loneliness on depressive symptoms, as well as the moderating role of social support in the relationship between internalized transphobia and psychological well-being. In this model, age, gender assigned at birth, actual gender identity, education level, stable relationship, community size, and religious education were included as control variables.

The moderated mediation model was tested using the PROCESS Macro (i.e., Model 5) [48] with bias-corrected bootstrapping (5000 samples) and 95% confidence intervals (CIs). Following Hayes’s recommendations, the indirect effect can be considered significant whether both the upper and lower boundaries of the bias-corrected 95%*CI* do not contain zero. All continuous variables were centered before performing the analyses.

## 4. Results

### 4.1. Descriptive Statistics and Bivariate Correlations

Means, standard deviations, ranges, and bivariate correlations between internalized transphobia, loneliness, social support, and psychological well-being are reported in Table 2.

The results show that all variables were correlated with each other. Specifically, internalized transphobia and loneliness correlated positively with each other and negatively with social support and psychological well-being. In contrast, the latter two variables were positively correlated with each other.

### 4.2. Direct and Indirect Effects of Internalized Transphobia and Loneliness on Psychological Well-Being

As shown in Figure 1, and in support of our first hypothesis, we found that internalized transphobia was positively associated with loneliness (*β* = 0.04, standard error (*SE*) = 0.01, 95%*CI* [0.02, 0.05], *p* < 0.001).

Similarly, in support of our second hypothesis, the results show that both internalized transphobia (*β* = −0.03, *SE* = 0.01, 95%*CI* [−0.04, −0.01], *p* = 0.001) and loneliness (*β* = −0.43, *SE* = 0.09, 95%*CI* [−0.63, −0.24], *p* < 0.001) were negatively associated with psychological well-being Figure 2.

When we included loneliness as a mediator, there was a significant overall effect (*β* = −0.04, *SE* = 0.01, 95%*CI* [−0.06, −0.03], *p* < 0.001), while the direct effect remained significant, indicating a case of partial mediation and confirming our third hypothesis. Indeed, the indirect effects showed that loneliness significantly mediated the association between internalized transphobia and psychological well-being (*β* = −0.02, *SE* = 0.01, 95%*CI* [−0.03, −0.01]). In addition, internalized transphobia and loneliness explained a significant proportion of the variance in psychological well-being (*F* = 9.37, *R*^2^ = 0.52, *p* < 0.001).

In addition, the control variables that showed a significant association with psychological well-being were age (*β* = 0.05, *SE* = 0.02, *p* = 0.04), educational level (*β* = −0.76, *SE* = 0.22, *p* = 0.001), and stable relationship (*β* = 0.45, *SE* = 0.15, *p* = 0.004), suggesting that individuals who were older, less educated, and in a stable relationship were more likely to report higher levels of psychological well-being than their counterparts.

### 4.3. The Moderating Role of Social Support

Regarding Hypothesis 4, social support proved to be a significant moderator (*b* = −0.06, *p* = 0.04), as the effect of internalized transphobia on psychological well-being decreased with moderate (*b* = −0.03, 95%*CI* [−0.04, −0.01], *p* = 0.001) and high (*b* = −0.04, 95%*CI* [−0.06, −0.01], *p* < 0.001), but not with low (*p* < 0.05), social support. These data seem to suggest that internalized transphobia has less impact on psychological well-being among those who feel they receive significant social support than among those who feel they do not receive adequate social support.

The graphical interaction plot is shown in Figure 3.

## 5. Discussion

The current study examined the relationships between internalized transphobia, loneliness, social support, and psychological well-being in a group of Italian TGNC individuals. The main results showed that all variables were correlated with each other. Furthermore, it emerged that loneliness partly mediated the relationship between internalized transphobia and psychological well-being. Social support also emerged as a significant moderator, as the impact of internalized transphobia on psychological well-being decreased at moderate social levels and high support levels, but not at low levels.

In the Italian context, the gaps relating to the topic addressed are significant; these results help to identify the risk and protection factors for the well-being of TGNC youth in the country.

We found that internalized transphobia was negatively associated with psychological well-being. This finding is consistent with findings in the literature [17,49,50], confirming that internalized transphobia is a significant risk factor for psychological well-being in TGNC populations. Specifically, this finding is particularly consistent with Hendricks and Testa’s [17] theories about the detrimental effects of internalized transphobia on the mental health of TGNC individuals, as well as internalized homophobia for the lesbian, gay, and bisexual population.

Furthermore, we highlighted that loneliness was a significant mediator between internalized transphobia and psychological well-being. This is the most innovative finding of the current study, as it sheds light on possible mediators of a relationship generally considered to be direct (i.e., internalized transphobia and health outcomes) [22,50]. Thus, we can assume that one of the reasons internalized transphobia increases the likelihood of developing negative mental health outcomes is because this proximal stressor creates the perception of being alone in a world that stigmatizes gender diversity, which likely also makes it difficult to benefit from the support of the TGNC community. Another possible explanation is that our sample consists of TGNC youth who are likely still developing their transgender identity, which is not as normative as cisgender identity, so they may feel disconnected from the collective identity and thus feel alone [21,51].

In fact, cisgender and heterosexual people can experience a potentially harmonious development of sexual identity since it does not conflict with social and cultural standards. Instead, in general, young LGBT+ people may face various complications due to the interference of family, social, cultural, religious, peer group, and institutional pressures. The development of a stigmatized sexual identity, in fact, must cause a struggle to preserve a coherent and authentic sense of self [52,53].

Research has been undertaken to investigate these specificities, yet significant empirical gaps remain regarding ways to support people who identify as a sexual minority during the period of sexual identity development. Understanding the best strategies to foster healthy development would allow us to orient it towards an identity experienced as positive and an integrated self-concept [54].

However, these are only speculative, interpretive hypotheses that could benefit from a qualitative approach that delves into the psychodynamics of such processes.

Finally, regarding the moderating role of social support, our findings confirm a long tradition of studies highlighting that social networks are one of the best protective factors against negative health outcomes [55,56,57,58,59].

It is necessary to emphasize the importance of the latter aspect also in relation to the age of the participants in the research, who have recent experiences of peer networking in contexts such as school and university. For example, a good support network can be built in contexts such as school but, in Italy, there is a discontinuous presence of psychologists, despite the availability in the area of professionals with adequate experience and training [60,61]. If, therefore, it becomes conceivable to create a professional system that not only must not be conditioned by the initiatives of the individual school or by specific conventions, but that is also capable of counteracting, from a systemic perspective, the negative impacts of heteronormativity, sexism, and transphobic bullying [62,63].

## 6. Limitations and Future Research

The results of this study provide important insights into the well-being of young Italian TGNC who, as stated above, represent an area not sufficiently explored by the various national scientific disciplines. Furthermore, it is important to highlight some significant limitation in the interpretation of the results. First, the online recruitment of participants, although facilitating the search for some aspect, does not guarantee the representativeness of the experiences of TGNC individuals (internet access limitations for some subjects). Second, the cross-sectional nature of the study does not allow for certainty about the directionality of the variables, which would require longitudinal studies. Third, the participants in the research are almost all Caucasians and this does not allow for any differences related to the ethnicity of TGNC people, since people of colors suffer high rates of discrimination compared to Caucasians [64].

Finally, the sample was relatively small, so it could be split into two macro categories (binary vs. non-binary) only for reasons of statistical parsimony. Future studies should strive to recruit a larger sample and not reduce participants to two categories, which may not be representative of the complexity of TGNC community.

The results that emerged can, in our opinion, encourage the involvement of a larger group of participants, including through the use of pencil and paper, both to overcome the obstacles of participating online and to capture differences related to different socio-demographic variables. Furthermore, the development of the study could be based on longitudinal research to better identify the protective factors of the psychological well-being of the TGNC community based on age.

## 7. Conclusions

The study focuses on TGNC young people who, in Italy, do not enjoy significant visibility. We wanted to highlight the impact of stigma on the well-being of young people who participated in the survey, and who are part of gender minorities, as well as the role, respectively, of internalized transphobia, negatively associated with individual well-being, and also of loneliness and social support.

Thus, based on our findings, we can argue that it is critical to both expand the social network by fostering intimate and collaborative relationships and, at the clinical level, to manage the experiences of loneliness, which can also become radicalized over time as a result of early experiences in personal and family history.

These are interesting results for the Italian territory, which, as a host country, registers a massive presence of migrants, who also include young LGBT people who are particularly exposed to the risk of social and family exclusion due to the values of the country of origin, which, in some cases, carries the death sentence for LGBT people. In this sense, the influence of the first generation on the formation of children’s attitudes towards LGBT people can also push the second generation into invisibility and the related discomfort, or into the risk of loneliness, physical violence, and death [65].

Finally, regarding clinical implications, our findings should be considered in clinical interventions, as most TGNC mental health problems are socially determined [66,67,68,69,70], and working with the TGNC community to build significant social networks can mitigate the negative effects of stigma on health. According to Coleman et al. [71]’s recent theories of stigma [72,73] and the World Health Organization’s [74] call to consider gender differences, the well-being and health of TGNC individuals depend not only on appropriate clinical support [75], but also on a social context that is able to recognize citizenship and equality in the face of social stigma in many cultures [64,76].

## Figures and Tables

**Figure 1 healthcare-10-02282-f001:**
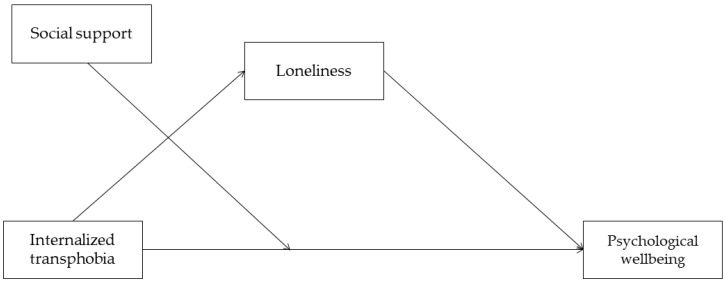
The hypothesized moderated mediation model.

**Figure 2 healthcare-10-02282-f002:**
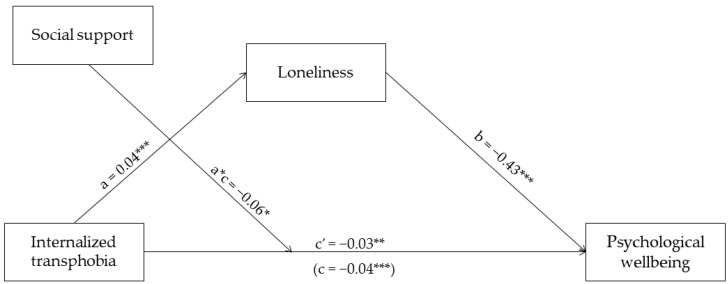
Results from the moderated mediation model. *** *p* < 0.001, ** *p* < 0.01, * *p* < 0.05.

**Figure 3 healthcare-10-02282-f003:**
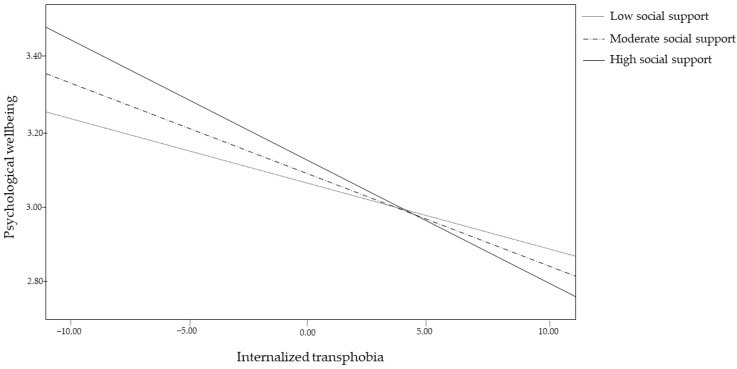
Interaction effect of internalized transphobia by social support on psychological well-being.

**Table 1 healthcare-10-02282-t001:** Socio-demographic characteristics of participants (*N* = 79).

Variable	*n* (%) or *M* ± *SD*
Age	23.73 ± 3.59
Range	18–30
Sex assigned at birth	
Male	22 (27.8)
Female	57 (72.2)
Actual gender identity	
Binary	45 (57)
Non-binary	34 (43)
Education	
≤high school	8 (10.1)
≥college	71 (89.9)
Ethnicity	
Caucasian	76 (96.2)
Non-Caucasian	3 (3.8)
Stable relationship	
Yes	38 (48.1)
No	41 (51.9)
Community size	
Urban	64 (81)
Non-urban	15 (19)
Religious education	
Yes	65 (82.3)
No	14 (17.7)

Notes: *M* = Mean; *SD* = Standard Deviation.

**Table 2 healthcare-10-02282-t002:** Correlations between internalized transphobia, loneliness, social support, and psychological well-being.

Scales	1	2	3	4	M (SD)	Range
1. Internalized transphobia	-				18.00 (8.66)	8–40
2. Loneliness	0.43 ***	-			2.23 (0.69)	1–4
3. Social support	−0.32 **	−0.76 ***	-		4.85 (1.19)	1–7
4. Psychological well-being	−0.53 ***	−0.62 ***	0.48 ***	-	3.08 (0.77)	1–5

Notes: *M* = Mean; *SD* = Standard Deviation. ** *p* < 0.01; *** *p* < 0.001.

## Data Availability

The datasets generated during the current study are available from the corresponding author on reasonable request.

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
