# Peer review of "The Role of Internalized Transphobia, Loneliness, and Social Support in the Psychological Well-Being of a Group of Italian Transgender and Gender Non-Conforming Youths"

_healthcare, 2022, doi:10.3390/healthcare10112282_

Round 1
Reviewer 1 Report
I have reviewed the manuscript entitled “The role of internalized transphobia, loneliness, and social support in the psychological well-being of a group of Italian transgender and gender non-conforming youths.” I think that the paper is about an important topic. Below, there are some minor suggestions that I think will improve the quality of the paper.
I can say that I have read an Introduction that summarizes the background of this paper quite nicely.
More detailed information should be given about how the participants were included in the study. "The link was posted on the website of social and educational institutions, self-help associations, and health organizations." Which websites are meant here? What is shared text on the site? In which sentence(s) were the participants invited? This needs to be specified.
In addition, it would be appropriate to briefly summarize the main findings in the first paragraph. From the second paragraph, the findings can be discussed. In the discussion, the findings should not be expressed as "first" , "second" and “third”.
The third finding should be discussed.
Please write a paragraph containing the limitations of this study.
Reviewer 2 Report
Thank you so much for the opportunity to read this manuscript. It was great to see an article that challenges people to understand the relationship between loneliness, social support, as it relates to the psychological well-being of TGNC youth. I believe that this article has much to contribute. Below you will find my comments concerning your manuscript.
· Introduction
o In your introduction, you do a good job of articulating the wide diversity of TGNC people and how they are bound together due to the oppression they experience. However, I was hesitant at your usage of “heteronormative cultural rules,” because I would argue it’s more trans oppressive cultural rules than heteronormativity (though for some, heteronormativity may also compound their oppression).
o What I would encourage you to further emphasize in the introduction is how environments and systems have to shift in regard to these realities of internalized transphobia or loneliness. Right now, you run the risk of presenting these youth in very deficit views instead of emphasizing how these are connected to problematic contexts. Make this explicit, especially as you introduce your current study. What do audiences/readers have to gain from this understanding, especially as it relates these systems?
· Methods
o Note that your table has “Gender assigned at birth.” Shouldn’t this be sex assigned at birth?
o Trans scholars have offered critiques about presenting the trans community in binary/non-binary ways (https://medium.com/national-center-for-institutional-diversity/not-another-gender-binary-a-call-for-complexity-over-cis-readability-d9eaefdcefc2). I would encourage you to think through whether this is an appropriate way to describe your participants.
o I would argue that you would need a justification for using the loneliness scale that you did given that it was intended for children and adolescents–but your project included ages up to 30, which is outside of this range.
· Discussion
o Within your discussion, I would like to see more attention to the Italian context in which your study was situated. What does the Italian context mean for your work?
o I would also like to see you unpack the idea that developing trans identity is not as normative as developing a cis identity–especially given my concern about that you’re presenting TGNC communities in a deficit light. Implicate the systems and structures, not the communities.
· Conclusion
o When you articulate the implications of your work, it would be more compelling to articulate what these implications would look like specifically in the Italian context.
